# HIGHLIGHT: LEARNING VISUAL PROMPTS FOR VISION-LANGUAGE MODELS

## ABSTRACT

Large-scale Vision-Language Models, such as CLIP, demonstrate impressive capabilities and have multiple applications, from text-to-image generation to zero-shot classification. Recent work has suggested that visual prompts, such as a red circle, can steer the vision encoder to the circled region. While such vision prompts have now been used in various applications, they might be model-specific and depend on the model learning these behaviours from its training data. Discovering and evaluating various prompts might not be feasible given different models, tasks, and datasets. In this paper, we propose *Highlight*, a method to learn a visual prompt that highlights a region in an image or refines a manually engineered visual prompt. Using our framework, we can learn to highlight in a supervised way using a dataset of text-image region pairs or in an unsupervised way using synthetic captions or images only. *Highlight* outperforms other visual prompts, prompt learning approaches, and compute-intensive methods that use ensembles of multiple models and visual prompts.

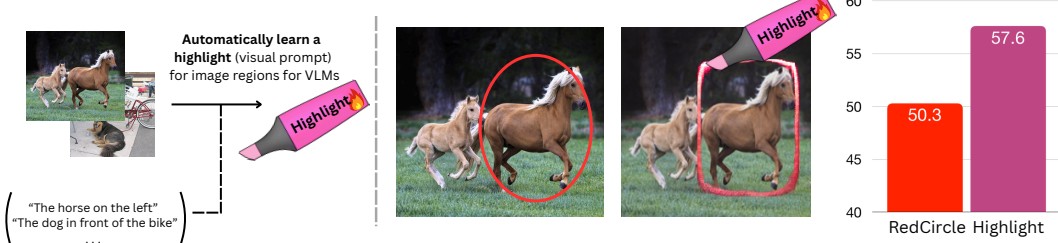

Figure 1: **Highlight automatically learns to visually prompt images.** Left: Given an image collection and, optionally, descriptions for image regions, Highlight learns a visual marker that is salient for VLMs. Right: A red circle vs the visual marker learnt by Highlight. Highlight outperforms a red circle by 15% on the RefCOCO, RefCOCO+ and RefCOCOg datasets, on average.

## 1 INTRODUCTION

Vision Language Models (VLMs) have shown outstanding performance across a range of tasks: For instance, CLIP (Radford et al., 2021) can perform zero-shot classification in a number of domains, and Flamingo (Alayrac et al., 2022) can count objects within an image, even though these tasks were not explicitly part of their training. Instead, it is hypothesised that these abilities *emerge* from the diverse large-scale training data. However, since the alt texts in pretraining image-text datasets summarise global image descriptions, these models have limited capabilities for localising objects. This limited spatial awareness hinders the usage of VLMs in various potential applications, such as autonomous driving (Wen et al., 2023), robotics (Brohan et al., 2023; Driess et al., 2023; Gao et al., 2023), and assistive technologies (Yang et al., 2022).

A promising new approach, *visual prompting* (Wu et al., 2024), has emerged to address these limitations. By introducing visual markers, such as red circles, to highlight target objects, pretrained foundation models can reason about localised image regions. Visual prompting for VLMs has found

multiple applications, such as medical image analysis (Denner et al., 2024), trajectory selection (Song et al., 2024), and audio descriptions generation Xie et al. (2024b). However, despite the different domains of application of visual prompting and the different visual encoders used, these methods rely on simple markers, such as a red circle, which have only been found to perform well empirically.

It has been suggested that VLMs learn to pay attention to such visual markers from their training data, *e.g.* Shtedritski et al. (2023) discover images annotated with red circles in YFCC100M (Thomee et al., 2016), which is part of CLIP's pretraining data. However, whether a red circle is the prompt that best elicits these emergent behaviours in VLMs is unclear. Furthermore, there might not be one optimal prompt per model—different domains in the training data might be annotated differently because of arbitrary conventions in fields like medicine or science. Thus, it becomes unfeasible to manually guess and evaluate visual prompts for every new VLM and domain to which it is applied.

In this paper, we introduce *Highlight*, a method to automatically learn a visual prompt that highlights regions for a VLM. Highlight visually prompts an image in a differentiable way by predicting both the shape and the colour of the visual marker. To supervise Highlight, first, we consider the case when annotated text-image region pairs are available. We show that in that case, we can use CLIP's multimodal contrastive objective to match prompted images with text, learning a state-of-the-art visual marker. However, this requires manual annotations. We thus introduce an unsupervised version of Highlight that is trained with images only and does not require manual supervision. Our key insight is that we can replace the textual description of a region in the image-text contrastive objective either with a synthetically generated caption for the image patch or with a visual description of the region, *e.g.* a crop. For example, we can learn to match highlighted images with a crop of the image. We find that the visual marker we learn in that way outperforms manually engineered visual markers.

## 2 RELATED WORK

**Visual prompt enginering**, as introduced by Shtedritski et al. (2023), uses visual markers in the input image to change the output of a VLM. For example, Shtedritski et al. (2023) draw red circles around objects of interest in order to make the VLM focus on these specific areas, whereas Yang et al. (2024) propose pixel-wise outlines of objects in the image. (Xie et al., 2024a) proposes to optimise the position of a visual marker in order to discover objects within an image. Both Yang et al. (2024) and Shtedritski et al. (2023) hypothesise that their proposed prompts are salient for the model because similar visual markers are present in the training data. Instead, Rezaei et al. (2024) optimise the colours within a fixed mask (*e.g.* a circle) to draw the attention of transformers towards the prompted area. However, the method of Rezaei et al. (2024) only applies to transformers and requires access to the internal features of the ViT. Instead, Highlight treats the vision model as a black box and is thus architecture-agnostic. Furthermore, our framework can accommodate various forms of supervision, such as using image-text pairs, which allows learning a *supervised* visual prompt.

**Referring Expression Comprehension** (REC) is the task of localising an object in an image corresponding to a query textual description. Many REC methods start with object proposals, for example, generated with Faster-RCNN (Ren et al., 2015), and then learn to score them (Hu et al., 2016; Luo & Shakhnarovich, 2017; Liu et al., 2020; Yang et al., 2019; Wang et al., 2019). Past research focuses on models specifically trained to solve the REC task. This includes combining pretrained vision and language models (Mao et al., 2016; Yu et al., 2016; Nagaraja et al., 2016; Hu et al., 2016; Luo & Shakhnarovich, 2017), learning a joint-embedding space (Wang et al., 2016; Liu et al., 2017; Chen et al., 2017), graph-based models (Liu et al., 2020; Wang et al., 2019; Yang et al., 2019), that yield better interpretability, and more recently transformer based architectures (Kamath et al., 2021; Kim et al., 2021; Deng et al., 2021). However, most of these rely on large amounts of training data, which include images, referring expressions, and the corresponding bounding boxes.

**Unsupervised Referring Expression Comprehension** has recently been enabled with the introduction of large pretrained models, such as CLIP (Radford et al., 2021), which allow solving REC in a *zero-shot manner* (Shtedritski et al., 2023; Yao et al., 2024; Subramanian et al., 2022; Kirillov et al., 2023b). ReCLIP (Subramanian et al., 2022) ranks object crops using CLIP before a post-processing step that considers relations between objects. CPT (Yao et al., 2021) colours objects and uses a pre-trained captioning model (Zhang et al., 2021) to predict which colour corresponds to the referring expression. More recently, unsupervised methods overlay visual prompts on objects (Shtedritski et al., 2023; Yang et al., 2024; Rezaei et al., 2024). However, most of these methods use ensembles

of visual prompts and/or vision-language models to achieve competitive performance. Instead, our method automatically discovers a visual prompt that outperforms such ensembles, making it several times more compute-efficient.

**Visual patches** can be optimised to steer the output of a vision model. Papernot et al. (2016) demonstrate that altering only a few pixel values can divert the model's output towards an adversarial output. Schlarmann & Hein (2023) recently transferred this work to VLMs, where the objective is to move the embedding of an adversarial image towards some harmful textual embedding. Their work differs from ours in requiring the adversarial changes to be invisible to humans. More closely related to our work, *adversarial patches* (Brown et al., 2017) restrict the alteration to a region. Our method can be seen as an extension of this research. However, unlike adversarial patches, we do not aim to mislead the model's output towards a harmful or incorrect outcome. Instead, we focus on optimising visual prompts to improve task performance.

## 3 METHOD

We aim to learn a *visual prompt* for a VLM that focuses the visual representation on a particular object in an image. To solve a prediction task, models such as CLIP jointly encode images and text and learn to match them. An input to such a model is an image $I \in \mathbb{R}^{3 \times H \times W}$ and text $T$, and the output measures similarity between both inputs.

### 3.1 VISUAL PROMPTING

A model such as CLIP can be used to find an object referred to by a query text prompt $T$ by visually prompting an image. First, a visual prompting function $\mathcal{F}(I, u)$ is used to highlight one object proposal $u$ in the image $I$, where $u$ is the location of an object in the image, which can be, *e.g.* bounding box coordinates $u = (x_c, y_c, x_w, y_h)$. For a set of possible object locations $U$, the location $\hat{u}$ in the image that is best described by the referring expression $T$ is then given by

$$\hat{u} = \arg\max_{u \in U} s(\mathcal{F}(I, u), T), \tag{1}$$

where $s(I, T)$ is the compatibility score between an image and text, *e.g.* as measured by a VLM. To implement the visual prompting function $\mathcal{F}$, prior work crop the image (Subramanian et al., 2022), overlay red circles (Shtedritski et al., 2023), or generate pixel-wise outlines (Yang et al., 2024). In this section, we show how we can learn such a prompt.

### 3.2 OVERLAYING A VISUAL PROMPT.

We decompose a visual prompt as $\mathcal{F} = \mathcal{F}_\alpha \otimes \mathcal{F}_{RGB}$, where $\mathcal{F}_\alpha \in \mathbb{R}^{H \times W}$ is an alpha channel and $\mathcal{F}_{RGB} \in \mathbb{R}^{3 \times H \times W}$ is a prompting image. To visually prompt an image $I$, we then have

$$\mathcal{F}(I, u) = \begin{cases} \mathcal{F}_\alpha(v) * \mathcal{F}_{RGB}(v) + (1 - \mathcal{F}_\alpha(v)) * I(v) & \text{if } v \in u, \\ i(v) & \text{else,} \end{cases} \tag{2}$$

performing alpha blending for all locations $v \in \mathbb{R}^2$ that are within the region $u$.

### 3.3 HIGHLIGHT: LEARNING A VISUAL PROMPT

Instead of using a manually engineered visual prompt, our method Highlight learns a visual prompting function $\mathcal{F}$ to highlight an object. A schematic of Highlight is shown in Fig. 2

**Learning a visual prompt.** We notice that the region selection in Eq. (1) resembles classification, where the most compatible region to the text query is selected. Given a dataset which consists of text queries $T$, where for each $T$ we have a positive and several negative regions (that are *not* described by $T$), $u$ and $U_n$ in an image $I$, we can directly learn $\mathcal{F}$ such that we fulfill Eq. (1):

$$\mathcal{L}_s(I, u, U_n, T) = \mathcal{L}_{\text{InfoNCE}}(T, \mathcal{F}(I, u), \mathcal{F}(I, u_n)_{u_n \in U_n}), \tag{3}$$

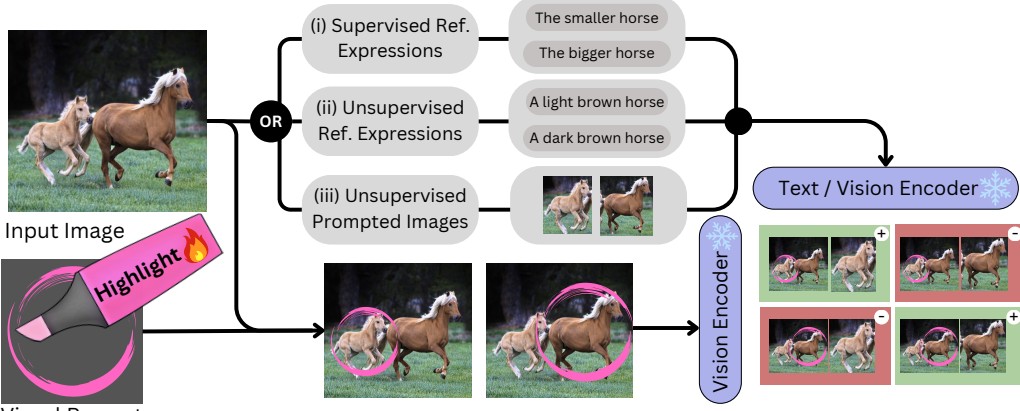

Figure 2: **Overview of Highlight.** Highlight generates a visual prompt, which is then alpha-blended with object proposals in the image. To learn the visual prompt, we construct positive and negative pairs for each object using: (i) supervised text and bounding box pairs (e.g. from RefCOCO), (ii) unsupervised text and bounding box pairs obtained by captioning the bounding box, (iii) a visual representation of the object, *e.g.* a crop of the bounding box as in the Figure, or the original image, visually prompted with a red circle. Here, (ii) and (iii) are unsupervised in that they do not require manual text-image region annotations. The CLIP image and text encoder are kept frozen.

where

$$\mathcal{L}_{\text{InfoNCE}}(x, x^+, X^-) = -\log\left(\frac{\exp\left(\text{sim}(x, x^+)/\tau\right)}{\exp\left(\text{sim}(x, x^+)/\tau\right) + \sum_{x^- \in X^-} \exp\left(\text{sim}(x, x^-)/\tau\right)}\right) \quad (4)$$

is the InfoNCE loss (Oord et al., 2018). Here, $\tau$ is a temperature parameter, and

$$\text{sim}(I, T) = \frac{\langle \Phi_I(I), \Phi_T(T) \rangle}{\|\Phi_I(I)\|\|\Phi_T(T)\|} \quad (5)$$

is the cosine similarity between the embeddings of the image and text using the CLIP image and text encoders, $\Phi_I$ and $\Phi_T$. Intuitively, minimising Eq. (3) will give us a prompting function $\mathcal{F}$ that focuses the global image embedding on the highlighted prompted object, making the image's embedding similar to the textual description of the prompted object while also making the embedding dissimilar to other textual descriptions of other objects in the image.

**Unsupervised visual prompts with synthetic text-image pairs.** The loss $\mathcal{L}_s$ in Eq. (3) is supervised in that it requires manual annotations in the form of region-specific captions, which can be limiting and expensive to acquire. Thus, we can augment this method by using synthetic captions $\tilde{T}_u$ (as recently used to train CLIP (Nguyen et al., 2024; Hammoud et al., 2024)) generated by BLIP-2 (Li et al., 2023) for a given crop. With this, we can also generate captions for the negative locations, giving us additional negative image-text pairs $T_{u_n}$. The unsupervised loss becomes:

$$\mathcal{L}_u(I, u, U_n, \tilde{T}) = \mathcal{L}_{\text{InfoNCE}}(\mathcal{F}(I, u), \tilde{T}_u, \tilde{T}_{u_n, u_n \in U_n}) + \mathcal{L}_{\text{InfoNCE}}(\tilde{T}_u, \mathcal{F}(I, u), \mathcal{F}(I, u_n)_{u_n \in U_n}) \quad (6)$$

**Unsupervised visual prompts with image-image pairs.** The naive, unsupervised objective proposed above suffers from two main drawbacks. First, obtaining the captions requires a captioning VLM, which might not be available for some domains and introduces additional computational overhead. Furthermore, the descriptions generated by the captioner might be *ambiguous*. To see why that is the case, consider the example in Fig. 2. The supervised referring expressions in datasets such as RefCOCO (Kazemzadeh et al., 2014) are explicitly designed such that the objects can be uniquely identified from their descriptions. However, this can not enforced on the captioning VLM, which has no global information about the scene. Instead of the captions {"A light brown horse" and "A dark brown horse"} (as in Fig. 2), the captioner might output {"A horse" and "A horse"}, from which the objects can not be disambiguated, leading to a sub-optimal learning signal.

Instead of matching a visually prompted image to a textual description of it, we propose to match it with the visual content in the prompt instead. To this end, we replace each description $T_u$ of the region $u$ in Eq. (6), with a visual representation of the region $u$. We denote this with $\mathcal{F}^*$, where $\mathcal{F}^*(I, u)$ might be any visual prompting function, such as a crop of the region $u$ or the image, prompted with a red circle in $u$. This gives us the unsupervised objective:

$$
\begin{aligned}
\mathcal{L}_{u,i2i}(I, u, U_n) = &\mathcal{L}_{\text{InfoNCE}}(\mathcal{F}(I, u), \mathcal{F}^*(I, u), \mathcal{F}^*(I, u_n)_{u_n \in U_n}) \\
&+ \mathcal{L}_{\text{InfoNCE}}(\mathcal{F}^*(I, u), \mathcal{F}(I, u), \mathcal{F}(I, u_n)_{u_n \in U_n}).
\end{aligned}
\tag{7}
$$

**Visibility loss.** We empirically observe that learning the alpha channel is sometimes unstable and the model converges to outputting a zero alpha channel, which results in poor performance. To tackle this, we simply penalise low average values of $\mathcal{F}_\alpha$:

$$
\mathcal{L}_v = \max(0, \tau - \frac{1}{HW} \sum_{x,y} \mathcal{F}_\alpha(x, y)),
\tag{8}
$$

where $\tau$ is a threshold which controls when the loss turns off.

**Training formulation.** We freeze the CLIP text and image encoders and only train the prompt function $\mathcal{F}$, which we parametrise with a neural network. Given a dataset $\mathcal{D}$ of images and object proposals, our loss is

$$
\mathcal{L} = \frac{1}{|\mathcal{D}|} \sum_{i, u, U_n \in \mathcal{D}} \beta \mathcal{L}_u(i, u, U_n) + \gamma \mathcal{L}_v,
$$

where $\beta$ and $\gamma$ are hyperparameters. In practice, given a set of object proposals $U$, we can select any $u \in U$ and define the set of negative proposals as $U_n = U \setminus u$. If region-text pairs are available, we can swap $\mathcal{L}_u(i, u, U_n)$ for the supervised loss $\mathcal{L}_s(i, u, U_n, T)$.

**Pretraining** We found that our unsupervised method, especially when trained with image-image pairs, performs better when $\mathcal{F}$ has been initialised with a manually engineered visual marker, *e.g.* a red circle. Given such a marker, we pretrain $\mathcal{F}_{RGB}$ with mean squared error loss and $\mathcal{F}_\alpha$ with binary cross entropy loss.

## 4 EXPERIMENTS

In this section, we describe the datasets used for our experiments and the implementation details of our method. After that, we evaluate Highlight against competing methods and ablate the different components we propose.

### 4.1 DATASETS

We evaluate Highlight on Referring Expressions Comprehension. We use the RefCOCO (Yu et al., 2016), RefCOCO+ (Yu et al., 2016), and RefCOCOg (Mao et al., 2016) datasets, all of which consist of images from the MS-COCO dataset (Lin et al., 2014) together with expressions that refer to a *unique* object in the image, which are also annotated with a bounding box. The unique expression-image region correspondence satisfies our condition in Eq. (3), where locations outside the bounding must *not* be described by the referring expression. RefCOCO+ only contains appearance-based expressions, whereas RefCOCO and RefCOCOg contain relation-based expressions (e.g., containing the words left/closer/bigger). The test sets of RefCOCO and RefCOCO+ are split into two parts, where "TestA" and "TestB" contain only people and non-people, respectively. Following prior work (Subramanian et al., 2022; Yao et al., 2021; Shtedritski et al., 2023), we use the bounding box proposals of MAttNet (Yu et al., 2018). For our unsupervised losses, we remove proposals with bounding box sizes of less than 50px. We train on all RefCOCO datasets combined, making sure to exclude any images that appear in the test sets (e.g. we found that some training images of RefCOCOg might appear as test images in RefCOCO+). Following prior work, for evaluation we exclude test samples that have no correct proposals.

Table 1: **Performance on Referring Expressions Comprehension.** Our supervised prompt outperforms prior works that use ensembles of several models and prompts, which require up to 6x more forward passes. Our unsupervised prompt outperforms all other single-prompts methods by 6.2% on average. † as reported in (Rezaei et al., 2024). ‡ as reported in (Shtedritski et al., 2023). ⚥ contains expressions referring to people, ⚲ for objects. **Sup** indicates whether a method was supervised or not. The best result is marked in **bold**.

| Method | Sup | Ensemble | RefCOCO | | | RefCOCO+ | | | RefCOCOg | |
|---|---|---|---|---|---|---|---|---|---|---|
| | | | Val (⚥, ⚲) | TestA (⚥) | TestB (⚲) | Val (⚥, ⚲) | TestA (⚥) | TestB (⚲) | Test (⚥, ⚲) | Val (⚥, ⚲) |
| Crop | ✗ | ✗ | 31.6 | 30.9 | 36.8 | 36.8 | 36.0 | 41.0 | 53.8 | 55.4 |
| Reverse Blur | ✗ | ✗ | 36.5 | 40.5 | 36.7 | 41.4 | 45.2 | 38.6 | 49.6 | 49.9 |
| Red Circle (Shtedritski et al., 2023) | ✗ | ✗ | 45.1 | 50.1 | 41.6 | 50.1 | 54.8 | 47.5 | 56.8 | 56.6 |
| Guided Attention† (Rezaei et al., 2024) | ✗ | ✗ | 46.7 | 49.5 | 43.9 | 48.1 | 51.3 | 46.2 | 50.4 | 49.0 |
| Highlight (Unsupervised, synth captions) | ✗ | ✗ | 46.9 | 51.2 | **45.3** | 53.0 | 56.2 | **51.0** | 56.6 | **56.9** |
| Highlight (Unsupervised, images only) | ✗ | ✗ | **47.7** | **52.5** | **45.3** | 53.6 | 58.1 | 50.6 | **56.9** | 56.9 |
| ReCLIP† (Subramanian et al., 2022) | ✗ | ✓ | 45.8 | 46.1 | 47.1 | 47.9 | 50.1 | 45.1 | 59.3 | 59.0 |
| RedCircle‡ (Shtedritski et al., 2023) | ✗ | ✓ | 49.8 | 58.6 | 39.9 | 55.3 | 63.9 | 45.4 | 59.4 | 58.9 |
| Highlight (Supervised) | ✓ | ✗ | **53.9** | **59.7** | **49.4** | **59.4** | **64.3** | **53.3** | **64.7** | **63.3** |

Table 2: **Evaluating the importance of optimising the prompt shape.** We fix the prompt shape of Highlight (unsupervised) to a circle and only learn its RGB values, which is fairly comparable to Guided Attention (Rezaei et al., 2024). † as reported in Rezaei et al. (2024). The best result is marked in **bold**, and the second best underlined.

| Method | Shape | RefCOCO | | | RefCOCO+ | | | RefCOCOg | | Avg |
|---|---|---|---|---|---|---|---|---|---|---|
| | | Val (⚥, ⚲) | TestA (⚥) | TestB (⚲) | Val (⚥, ⚲) | TestA (⚥) | TestB (⚲) | Test (⚥, ⚲) | Val (⚥, ⚲) | |
| Guided Attention† | circle | 46.7 | 49.5 | 43.9 | 48.1 | 51.3 | 46.2 | 50.4 | 49.0 | 48.1 |
| Highlight (RGB only) | circle | 43.0 | 48.0 | 41.1 | 49.6 | 53.3 | 47.0 | 54.3 | 53.3 | 48.7 |
| Highlight | learnt | **47.7** | **52.5** | **45.3** | **53.6** | **58.1** | **50.6** | **56.9** | **56.9** | **52.7** |

## 4.2 IMPLEMENTATION DETAILS

**Architecture.** While one can optimise a visual prompt directly in pixel space, the limited number of learnable parameters in pixel space hinders the sharing of spatial patterns between pixels, leading to less efficient optimisation (Krull et al., 2019). To generate the visual prompt, we, therefore, use a Coordinate Network (Stanley, 2007), which, given the $(x, y)$ coordinates of a pixel location in the visual prompt, generates the RGB and $\alpha$ values for that location. Therefore, Coordinate Networks can be thought of as 2D Nerfs (Mildenhall et al., 2021). The network generates a fixed-sized prompt, which is afterwards scaled to fit the size of the object proposal that should be prompted. We use a Coordinate Network with eight layers and a hidden dimension of 256. We add a sinusoidal positional encoding to the input and apply a sigmoid function to the outputted $\alpha$ mask.

**Additional details.** We use a linear annealing schedule with a 10% warmup and cooldown for all pretrained Highlights and pretrain them to output a red circle for 2,000 steps unless mentioned otherwise. For the models that were not pretrained, we empirically observed exploding gradients when utilising a scheduler. Therefore, we do not use a scheduler for these runs. Similarly to prior work, we use a CLIP ViT-L/14 in most experiments unless stated otherwise. The hyperparameters we used in different settings can be found in the Appendix.

## 4.3 RESULTS

**Supervised Highlight.** In Table 1, we compare Highlight to other visual prompting methods. Since our framework is the first to allow supervised training of a visual prompt, we compare to ensemble methods, which usually perform better than single prompt methods but have a significant inference overhead. Our supervised visual prompt outperforms the ensemble methods of ReCLIP, which uses post-processing with spatial relations and an ensemble of 2 models and 2 visual prompts for a total of 4 forward passes of the CLIP vision encoder, and RedCircle, which uses an ensemble of 2 models and 3 visual prompts for a total of 6 forward passes of a CLIP vision encoder. In comparison, the Highlight prompt achieves better performance, while only doing one pass of the CLIP vision encoder.

| Method | RefCOCO | RefCOCO+ | RefCOCOg |
|---|---|---|---|
| Crop | 33.1 | 37.9 | 54.6 |
| Highlight + Crop | 47.6 | 53.4 | 56.1 |
| Blur | 37.9 | 41.7 | 49.8 |
| Highlight + Blur | 48.5 | 54.1 | 56.9 |
| RedCircle | 45.6 | 50.8 | 56.7 |
| Highlight + RedCircle | 46.4 | 52.0 | 56.1 |

| Method | Pretraining | RefCOCO | RefCOCO+ | RefCOCOg |
|---|---|---|---|---|
| Highlight (supervised) | ✗ | 54.3 | 59.0 | 64.0 |
| Highlight (supervised) | ✓ | 51.5 | 56.3 | 59.7 |
| Highlight (unsup, synth captions) | ✗ | 47.1 | 52.5 | 55.5 |
| Highlight (unsup, synth captions) | ✓ | 47.8 | 53.4 | 56.8 |
| Highlight (unsup, images only) | ✗ | 39.2 | 45.5 | 49.3 |
| Highlight (unsup, images only) | ✓ | 48.5 | 54.1 | 56.9 |

Table 3: **Comparing different targets $\mathcal{F}^*$.** We compare different baseline visual prompts, with Highlight when trained with that visual prompt for its target $\mathcal{F}^*$. We see the biggest gains when using Crop, but the best performance when using Reverse Blur.

Table 4: **Pretraining Highlight improves performance.** We find that pretraining Highlight to output a visual prompt (red circle) leads to better performance in the unsupervised regime and the biggest gains are seen for the images-only model.

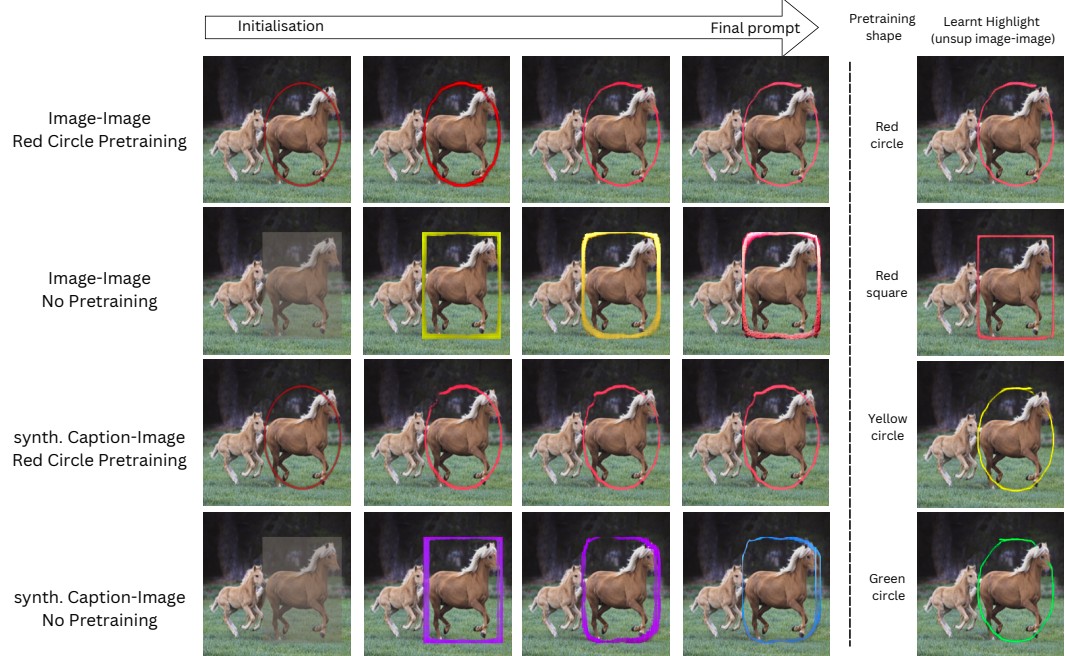

Figure 3: **Learnt visual markers.** Left: we show how the learnt visual markers in different training regimes change during training. We see that when no pretraining is used, Highlight converges to an outline of the prompted region, which is similar to the manually engineered prompts. Right: we show the final learnt prompts with several different pretraining visual markers.

**Unsupervised Highlight.** In Table 1, we also see the performance of the unsupervised version of Highlight in two settings—using synthetic captions and images only. We outperform other visual prompts on all REC datasets, including the concurrent work of Rezaei et al. (2024), which also learns a visual prompt from data. We see the biggest gains on the splits that contain only objects (non-humans), which we attribute to the fact that most regions in the dataset contain non-human objects. Additionally, the Highlight prompt learnt with images-only outperforms the one learnt with synthetic captions. Despite the modality gap (Liang et al., 2022) when training without text, our images-only framework is not affected by ambiguous synthetic captions, as discussed in Section 3.3.

**Prompt flexibility.** To evaluate the importance of learning the shape of the visual prompt, we fix the alpha channel $\mathcal{F}_\alpha$ to be a circle. In Table 2, we see that optimising $\mathcal{F}_\alpha$ significantly improves the learnt prompt. The restricted version of Highlight that uses a fixed circle mask can now be fairly compared to the concurrent work Guided Attention (GA) (Rezaei et al., 2024), who propose to learn a visual prompt by using a fixed circle shape whose colour is optimised. Similarly to Highlight, GA does that using images only but learns the RGB values of the prompt by focusing the attention of the CLIP ViT encoder on the prompted region. In Table 2, we see that Highlight's learning objective

Table 5: **Pretraining Highlight to output different shapes.** Here we show results for our unsupervised images-only model. We find that pretraining using a red circle leads to the best results. All pretrainings outperform a Highlight that was not pretrained at all.

| Pretrain Shape | RefCOCO | | | RefCOCO+ | | | RefCOCOg | |
|---|---|---|---|---|---|---|---|---|
| | Val (♀, ♻) | TestA (♀) | TestB (♻) | Val (♀, ♻) | TestA (♀) | TestB (♻) | Test (♀, ♻) | Val (♀, ♻) |
| Red Circle | **47.7** | **52.5** | 45.3 | **53.6** | **58.1** | **50.6** | **56.9** | **56.9** |
| Red Square | 47.0 | 50.6 | **46.0** | 52.3 | 55.7 | 48.4 | 55.3 | 54.8 |
| Green Circle | 47.2 | 52.1 | 45.3 | 53.0 | 57.6 | 50.2 | 55.7 | 56.0 |
| Yellow Circle | 45.1 | 49.4 | 43.2 | 50.7 | 54.9 | 48.0 | 55.1 | 54.2 |
| No Pretrain | 38.0 | 40.8 | 38.9 | 44.9 | 46.7 | 44.8 | 49.5 | 49.0 |

Table 6: **Results for different CLIP models**. We train Highlight in the three different modes, (i) unsupervised im2im, using image-image pairs only, (ii) unsupervised t2im, using synthetic text-image pairs, and (iii) supervised t2im, using ground truth text-image pairs. Overall, we see that pretraining improves performance in the unsupervised image-image regime, and does not help or hurts performance when either ground-truth or unsupervised captions are used.

| Model | Method | Supervision | Pretrain | RefCOCO | | | RefCOCO+ | | | RefCOCOg | |
|---|---|---|---|---|---|---|---|---|---|---|---|
| | | | | Val (♀, ♻) | TestA (♀) | TestB (♻) | Val (♀, ♻) | TestA (♀) | TestB (♻) | Test (♀, ♻) | Val (♀, ♻) |
| ViT-B/16 | Crop | — | — | 35.2 | 33.5 | 41.9 | 40.5 | 39.9 | 45.5 | 59.4 | 60.2 |
| | Blur | — | — | 40.2 | 42.0 | 40.1 | 43.9 | 48.0 | 43.6 | 55.7 | 55.8 |
| | Red Circle | — | — | 41.6 | 44.7 | 39.4 | 44.9 | 47.7 | 42.2 | 47.4 | 47.8 |
| | Highlight$_{im2im}$ | ✗ | ✓ | 44.9 | 47.6 | 41.9 | 48.8 | 51.7 | 44.7 | 49.6 | 49.9 |
| | Highlight$_{im2im}$ | ✗ | ✗ | 39.9 | 40.4 | 42.1 | 43.1 | 42.3 | 44.7 | 47.8 | 48.6 |
| | Highlight$_{t2im}$ | ✗ | ✓ | 43.6 | 46.6 | 40.3 | 47.4 | 50.8 | 43.7 | 48.6 | 48.9 |
| | Highlight$_{t2im}$ | ✗ | ✗ | 42.9 | 44.7 | 43.5 | 47.0 | 48.0 | 46.7 | 50.5 | 50.1 |
| | Highlight$_{t2im}$ | ✓ | ✓ | 50.8 | 56.1 | 46.7 | 53.9 | 60.0 | 49.0 | 55.8 | 55.9 |
| | Highlight$_{t2im}$ | ✓ | ✗ | 51.0 | 55.9 | 46.3 | 53.3 | 58.4 | 49.7 | 56.3 | 56.2 |
| ViT-L/14 | Crop | — | — | 31.6 | 30.9 | 36.8 | 36.8 | 36.0 | 41.0 | 53.8 | 55.4 |
| | Blur | — | — | 36.5 | 40.5 | 36.7 | 41.4 | 45.2 | 38.6 | 49.6 | 49.9 |
| | Red Circle | — | — | 45.1 | 50.1 | 41.6 | 50.1 | 54.8 | 47.5 | 56.8 | 56.6 |
| | Highlight$_{im2im}$ | ✗ | ✓ | 47.7 | 52.5 | 45.3 | 53.6 | 58.1 | 50.6 | 56.9 | 56.9 |
| | Highlight$_{im2im}$ | ✗ | ✗ | 38.0 | 40.8 | 38.9 | 44.9 | 46.7 | 44.8 | 49.5 | 49.0 |
| | Highlight$_{t2im}$ | ✗ | ✓ | 46.9 | 51.2 | 45.3 | 53.0 | 56.2 | 51.0 | 56.6 | 56.9 |
| | Highlight$_{t2im}$ | ✗ | ✗ | 45.5 | 49.9 | 45.8 | 52.0 | 54.6 | 51.0 | 55.6 | 55.4 |
| | Highlight$_{t2im}$ | ✓ | ✓ | 50.9 | 55.6 | 48.1 | 55.9 | 60.1 | 52.8 | 60.0 | 59.3 |
| | Highlight$_{t2im}$ | ✓ | ✗ | 53.9 | 59.7 | 49.4 | 59.4 | 64.3 | 53.3 | 64.7 | 63.3 |
| ViT-L/14@336 | Crop | — | — | 32.5 | 31.5 | 37.5 | 38.0 | 36.1 | 41.8 | 55.0 | 56.6 |
| | Blur | — | — | 35.4 | 38.4 | 36.0 | 40.8 | 43.5 | 38.3 | 49.8 | 49.7 |
| | Red Circle | — | — | 46.7 | 53.6 | 43.0 | 52.1 | 57.7 | 48.9 | 58.5 | 58.1 |
| | Highlight$_{im2im}$ | ✗ | ✓ | 46.7 | 52.1 | 43.2 | 52.1 | 57.8 | 48.4 | 56.8 | 56.6 |
| | Highlight$_{im2im}$ | ✗ | ✗ | 43.6 | 46.9 | 41.9 | 48.3 | 50.4 | 47.0 | 50.7 | 51.3 |
| | Highlight$_{t2im}$ | ✗ | ✓ | 47.7 | 53.7 | 43.8 | 52.7 | 58.0 | 49.3 | 56.7 | 57.4 |
| | Highlight$_{t2im}$ | ✗ | ✗ | 46.7 | 50.8 | 45.0 | 52.5 | 55.8 | 49.2 | 56.8 | 57.5 |
| | Highlight$_{t2im}$ | ✓ | ✓ | 53.6 | 60.8 | 47.3 | 58.9 | 65.4 | 53.3 | 62.3 | 62.3 |
| | Highlight$_{t2im}$ | ✓ | ✗ | 49.6 | 55.3 | 47.4 | 55.6 | 59.9 | 52.8 | 59.3 | 60.7 |

leads to a better visual prompt than the one learnt by GA. Furthermore, whereas GA (i) requires the vision encoder to be a ViT and (ii) uses the inner representation of the vision encoder, Highlight (i) is architecture agnostic, and (ii) treats the vision encoder as a black box and only uses its output.

**Effect of the target prompting function $\mathcal{F}^*$** We train Highlight with different target prompting functions $\mathcal{F}^*$ as part of the unsupervised image-image loss in Eq. (7). We see that learning Highlight using Blur (blurring everything outside the bounding box) leads to superior performance compared to training with a red circle prompt or with a crop. Also, using Crop as the target prompting function $\mathcal{F}^*$ leads to the biggest absolute gains over $\mathcal{F}^*$. We attribute this to the fact that we pretrain Highlight to output a red circle, which can then capture a complimentary signal from the crop/blur target functions. Still, we observe that the performance of the visual prompt learnt when the target $\mathcal{F}^*$ is a red circle outperforms using a plain red circle as the visual prompt.

**The importance of pretraining** Table 4 shows that pretraining Highlight to output a manually engineered visual prompt (red circle in this case) improves results in the unsupervised regime. As we decrease the level of supervision, the learning signal becomes weaker, and pretraining becomes more crucial, *i.e.* pretraining significantly improves performance when training only with images but has limited effects when training fully supervised. However, we notice that pretraining leads Highlight to a local minima as seen on the right of Fig. 3, where the final prompt has changed little compared to the initialisation. In Table 5, we show the performance of our image-only framework when pretraining

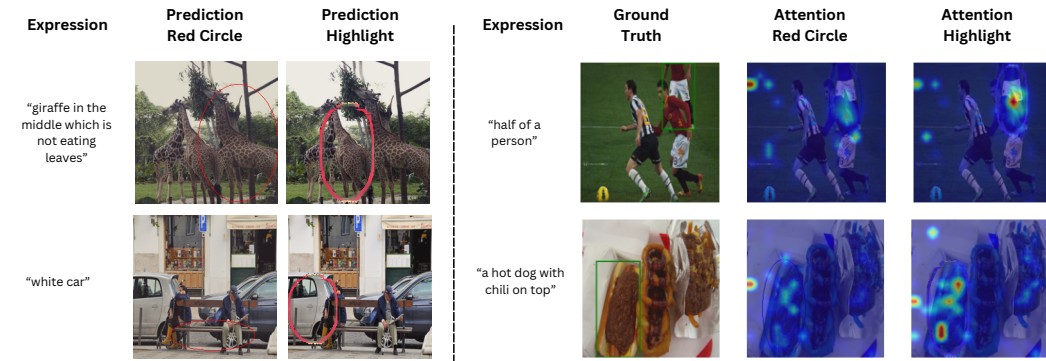

Figure 4: **Qualitative examples.** Left: We show examples of cases when supervised Highlight leads to a correct prediction on RefCOCO, but using a plain red circle fails at localising the correct region. Right: We show attention maps using the explainability method of Chefer et al. (2021). We see that when prompting with Highlight, we can successfully draw the attention of the visual transformer to the highlighted region.

with different visual prompts. We observe that Highlight, which is pretrained to output a red circle, outperforms the other pretraining shapes and colours. Moreover, we notice that both shape *and* colour of the pretraining prompt are important since changing the colour or changing the shape, *i.e.* from circle to square, leads to worse performance. Furthermore, we also notice that networks pretrained to produce better prompts tend to deliver better performance later on. For example, a red circle prompt performs better than a green circle prompt, which outperforms a yellow circle prompt (Shtedritski et al., 2023). This relationship remains consistent even after optimising each Highlight.

**Other CLIP models.** In Table 6, we show results when training Highlight on several CLIP models. Overall, we see that Highlight consistently improves over the performance of other visual prompts across models. For all models, the largest gains when pretraining are observed in the unsupervised image-image regime.

**Qualitative results.** In Fig. 3, we show how the visual prompts change during training, with different objectives and different pretraining. We see that the visual prompts that have been pretrained change little, and the subsequent training with our framework just refines them. On the right, we see this also holds when pretraining using other shapes and colours. The visual prompts learnt from scratch, however, resemble a contour around the object boundaries, which is similar to the manually engineered visual markers. This suggests that these emergent behaviours of VLMs, where they can be prompted with visual markers, are indeed learnt from data. In Fig. 4, we show attention maps generated using the method of Chefer et al. (2021). We see that Highlight can successfully steer the attention of the ViT to the highlighted object. Fig. 4 also shows examples where the predictions using a plain red circle fail, whereas Highlight correctly predicts the referred region.

## 5 CONCLUSION

Manually engineered visual prompts for VLMs have been extensively used for multiple tasks. However, choosing visual prompts requires trial and error and guessing. The framework we present, *Highlight*, automatically learns a visual prompt given a dataset of text and images, or images only. The optimised prompt can refine a manually engineered prompt, or be learnt from scratch, in which case we find they resemble manually engineered prompts. The visual prompts we automatically learn outperform manually engineered visual prompts, as well as computationally ensembles of models and prompts. We hope this inspires further work into automatically learning to highlight images, as well as applying such methods beyond CLIP models and Referring Expression Comprehension.

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

# A    APPENDIX

Table 7: **Results for heuristically adding spatial relations.** Adding spatial relations to the unsupervised captions improves performance—mainly on TestA splits.

| Method | RefCOCO | | | RefCOCO+ | | | RefCOCOg | |
|---|---|---|---|---|---|---|---|---|
| | Val (♀, ♟) | TestA (♀) | TestB (♟) | Val (♀, ♟) | TestA (♀) | TestB (♟) | Test (♀, ♟) | Val (♀, ♟) |
| Highlight (unsup, synth. captions) | 46.9 | 51.2 | 45.3 | 53.0 | 56.2 | 51.0 | 56.6 | 56.9 |
| Highlight (unsup, synth. captions) + SR | 46.9 | 51.8 | 45.3 | 53.0 | 57.0 | 50.7 | 56.7 | 58.0 |

## A.1    ADDING HEURISTICAL SPATIAL RELATIONS

As detailed in Section 3, one potential caveat of training with synthetic captions is that captions might be ambiguous, *e.g.* in Fig. 2 the caption model might output the same caption "a horse" for both object proposals in the image. We experiment with alleviating this ambiguity by introducing spatial relations using the heuristics introduced by Subramanian et al. (2022): For each object proposal, we identify the closest other object proposal and then determine which spatial relations hold between the two bounding boxes (*e.g.* left, right, above, below, smaller, bigger) and select one of these at random. Next, we combine the captions from the two objects with a randomly chosen synonym for the selected spatial relation. We take the list of synonyms from Subramanian et al. (2022). For example, in Fig. 2, this process might change the captions from {"a horse", "a horse"} to {"a horse to the left of a horse", "a horse bigger than a horse"}.

Table 7 shows the performance of training with synthetic captions both with and without heuristically adding spatial relations. Overall, performance when training with heuristically added spatial relations is slightly better than training without spatial relations. Interestingly, we observe that this improvement mainly stems from TestA splits. This might indicate that the captions the caption model generates are better suited to differentiate objects from one another than humans.

## A.2    TRANSFER BETWEEN MODELS

In Fig. 5, we see the relative performance improvement of Highlight over a network pretrained to output a red circle when trained on one CLIP model and evaluated on another. We see the biggest improvements when evaluating with the CLIP ViT-B/16 model and observe that the prompt optimised with CLIP ViT-L/14 is more transferable to other models than the other optimised prompts. Overall, a visual prompt learnt on a larger model transfers well on a smaller model, *e.g.* learning on ViT-L/14@336 and evaluating on ViT-B/16 work better than the opposite.

## A.3    COMPARISON TO FGVP

In Table 8 we compare Highlight to FGVP (Yang et al., 2024). FGVP is supervised in that it uses SAM (Kirillov et al., 2023a) to generate mask outlines for objects, which are used for visual prompts. Additionally, FGVP utilises several post-processing techniques. In Table 8, we see that unsupervised Highlight outperforms FGVP when it uses a single prompt. Supervised highlight outperforms FGVP when it uses up to 6 forward passes of the visual encoder and either of the two ad-hoc post-processing techniques. However, when FGVP uses 8 passes of the visual encoder and both post-processing methods, it performs better than Highlight. Overall, Highlight is a much more lightweight and efficient method, and performs favourably to FGVP.

## A.4    IMPLEMENTATION DETAILS

Table 9 gives an overview of the different hyperparameters for all runs. Note that if we train supervised, we iterate over referring expressions and corresponding bounding boxes instead of images. Therefore, we train for fewer but larger epochs. Moreover, we empirically observe that training in the supervised case leads to invisible prompts. Thus, we add a visibility loss as introduced in Eq. (8) with a threshold $\tau$ of 10%. For the pretrained Highlight$_{im2im}$ for the ViT-B/16 model, we use $\mathcal{F}^*$ crop instead of blur. To adjust to the higher resolution, we filter for object proposals with a minimum height and width of 75px for the ViT-L/14@336 model. Therefore, we consider the same

Table 8: **Comparison to FGVP.** We compare to FGVP (Yang et al., 2024), which is a visual prompting method, which is supervised in that it requires masks from SAM (Kirillov et al., 2023a). *SR* is the use of a spatial relations post-processing, that takes into account words like "left", "right", "bigger", "smaller", etc, which Subramanian et al. (2022) found to boost performance. *SS* is the use of Score Subtraction, which Shtedritski et al. (2023) found to boost performance. **Num Enc** is the number of forward passes of the visual encoder (number of prompts used times the number of visual encoders ensembled) ⚥ contains expressions referring to people, ⚲ for objects. **Sup** indicates whether a method was supervised or not. The best result is marked in **bold**, second best is underlined.

| Method | Sup | Num. Enc. | RefCOCO | | | RefCOCO+ | | | RefCOCOg | | Avg |
| --- | --- | --- | --- | --- | --- | --- | --- | --- | --- | --- | --- |
| | | | Val (⚥, ⚲) | TestA (⚥) | TestB (⚲) | Val (⚥, ⚲) | TestA (⚥) | TestB (⚲) | Test (⚥, ⚲) | Val (⚥, ⚲) | |
| Highlight (Unsupervised, synth captions) | ✗ | 1 | 46.9 | 51.2 | 45.3 | 53.0 | 56.2 | 51.0 | 56.6 | 56.9 | 52.1 |
| Highlight (Unsupervised, images only) | ✗ | 1 | 47.7 | 52.5 | 45.3 | 53.6 | 58.1 | 50.6 | 56.9 | 56.9 | 52.7 |
| FGVP | ✓ | 1 | 46.1 | 53.0 | 40.4 | 50.4 | 57.5 | 42.6 | 54.5 | 54.1 | 49.8 |
| FGVP$^{SR}$ | ✓ | 4 | 52.0 | 55.9 | 48.8 | 53.3 | 60.4 | 46.7 | 62.1 | 61.9 | 55.1 |
| FGVP$^{SS}$ | ✓ | 6 | 53.9 | 60.2 | 44.3 | 59.3 | 66.6 | 48.8 | 61.0 | 61.3 | 56.9 |
| FGVP$^{SR+SS}$ | ✓ | 8 | **59.6** | **65.0** | **52.0** | **60.0** | **66.8** | 49.7 | 63.3 | 63.4 | **60.0** |
| Highlight (Supervised) | ✓ | 1 | 53.9 | 59.7 | 49.4 | 59.4 | 64.3 | **53.3** | 64.7 | 63.3 | 58.5 |

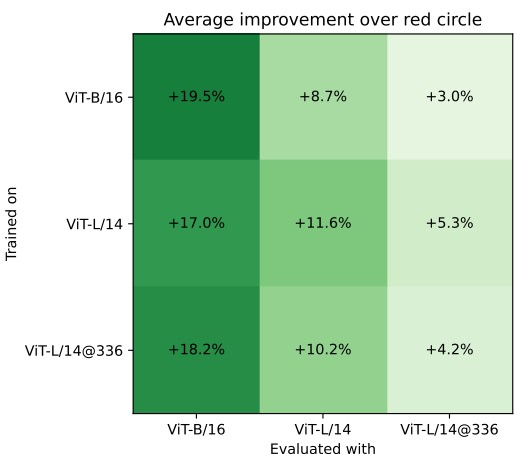

Figure 5: **Transferability of an optimised Highlight trained for one CLIP model to other models.** We use Highlights that were optimised with our image-image loss and pretrained to output a red circle for evaluation on other CLIP models. We observe that even though the Highlights were not optimised for that specific CLIP model, they still increase performance. We report the average performance increases across all RefCOCO test and validation datasets.

Table 9: **Hyperparameters for all optimised Highlights.**

| Model | Method | Sup | Pretrain | Learning Rate | Contrastive Loss Scaling $\beta$ | Visibility Loss Scaling $\gamma$ | Scheduler | Training Samples (Epochs) | Batch Size | Minibatch Size | Temperature |
| --- | --- | --- | --- | --- | --- | --- | --- | --- | --- | --- | --- |
| | Highlight$_{im2im}$ | ✗ | ✓ | 0.001 | 0.0001 | 0 | ✓ | ~100,000 (4) | 20 | 10 | 20 |
| | Highlight$_{im2im}$ | ✗ | ✗ | 0.001 | 0.0001 | 0 | ✗ | ~200,000 (8) | 56 | 15 | 20 |
| | Highlight$_{t2im}$ | ✗ | ✓ | 0.001 | 0.0001 | 0 | ✓ | ~100,000 (4) | 20 | 10 | 20 |
| ViT-B/16 | Highlight$_{t2im}$ | ✗ | ✗ | 0.001 | 0.0001 | 0 | ✗ | ~200,000 (8) | 20 | 10 | 20 |
| | Highlight$_{t2im}$ | ✓ | ✓ | 0.001 | 0.5 | 1.5 | ✓ | ~150,000 (1) | 16 | 25 | 20 |
| | Highlight$_{t2im}$ | ✓ | ✗ | 0.001 | 0.05 | 0.15 | ✗ | ~300,000 (2) | 16 | 25 | 20 |
| | Highlight$_{im2im}$ | ✗ | ✓ | 0.001 | 0.0001 | 0 | ✓ | ~100,000 (4) | 20 | 10 | 20 |
| | Highlight$_{im2im}$ | ✗ | ✗ | 0.001 | 0.0001 | 0 | ✗ | ~200,000 (8) | 20 | 10 | 20 |
| | Highlight$_{t2im}$ | ✗ | ✓ | 0.001 | 0.001 | 0 | ✓ | ~100,000 (4) | 20 | 10 | 20 |
| ViT-L/14 | Highlight$_{t2im}$ | ✗ | ✗ | 0.001 | 0.0001 | 0 | ✗ | ~200,000 (8) | 20 | 10 | 20 |
| | Highlight$_{t2im}$ | ✓ | ✓ | 0.001 | 0.5 | 1.5 | ✓ | ~150,000 (1) | 16 | 14 | 20 |
| | Highlight$_{t2im}$ | ✓ | ✗ | 0.001 | 0.05 | 0.15 | ✗ | ~600,000 (4) | 16 | 14 | 20 |
| | Highlight$_{im2im}$ | ✗ | ✓ | 0.001 | 0.00001 | 0 | ✓ | ~100,000 (4) | 20 | 10 | 20 |
| | Highlight$_{im2im}$ | ✗ | ✗ | 0.001 | 0.00001 | 0 | ✗ | ~25,000 (1) | 20 | 10 | 20 |
| | Highlight$_{t2im}$ | ✗ | ✓ | 0.001 | 0.0001 | 0 | ✓ | ~100,000 (4) | 20 | 10 | 20 |
| ViT-L/14@336 | Highlight$_{t2im}$ | ✗ | ✗ | 0.001 | 0.00001 | 0 | ✗ | ~100,000 (4) | 20 | 10 | 20 |
| | Highlight$_{t2im}$ | ✓ | ✓ | 0.001 | 0.1 | 0.3 | ✓ | ~150,000 (1) | 16 | 19 | 20 |
| | Highlight$_{t2im}$ | ✓ | ✗ | 0.001 | 0.05 | 0.15 | ✗ | ~150,000 (1) | 16 | 19 | 20 |

object proposals for both resolutions. We adjust batch size and minibatch size in order to fit on a single GPU, depending on the CLIP model and supervision regime used. Minibatch size denotes how many samples are considered at maximum for one iteration of the contrastive loss, *e.g.* if the minibatch size is 10, but there are 15 valid object proposals, they are cut down to 10 object proposals (at random), and for each positive pair in the contrastive loss, there are 9 negatives.

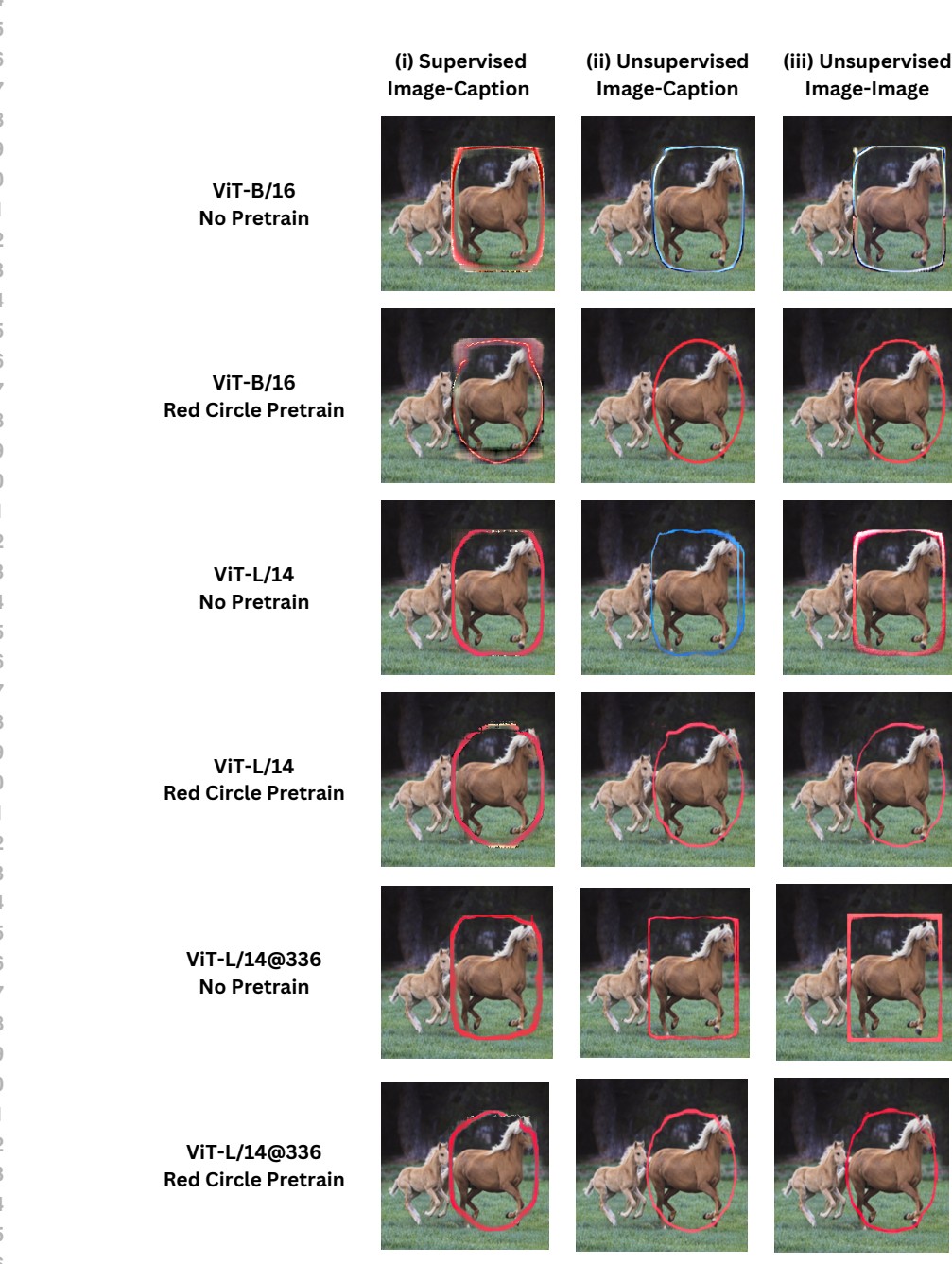

Figure 6: **Learnt visual markers.** We show the learnt visual marker for different CLIP models in the different supervision regimes.

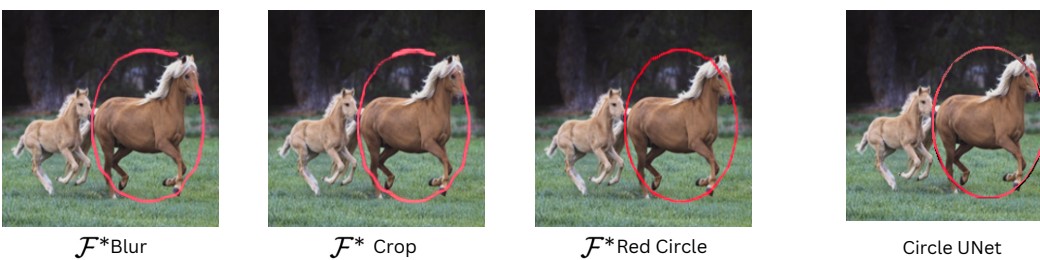

| $\mathcal{F}^*$Blur | $\mathcal{F}^*$ Crop | $\mathcal{F}^*$Red Circle | Circle UNet |

**Highlights optimized for different $\mathcal{F}^*$.** **Fixed mask.**

Figure 7: **Learnt visual prompts for ViT-L/14 trained with unsup. image-image loss.** Left: We show the learnt markers when optimising a Highlight pretrained with a red circle using different target functions. Right: We show the learnt prompt when training a Highlight with a restricted prompt shape to a circle. Similarly to Rezaei et al. (2024), we parametrise the prompt network with a UNet for this experiment.

## A.5 ADDITIONAL QUALITATIVE RESULTS

In Fig. 6, we show visual prompts we learn with Highlight for different CLIP models in all the supervision regimes we consider. We see that all models learn a contour in the unsupervised regime, where for the large models, this contour is red—resembling a red circle. In Fig. 7, we show (i) visual markers optimised for different target functions when using the unsupervised image-image objective and red circle pretraining and (ii) the learn visual marker when we compare to Rezaei et al. (2024), fixing the prompt shape to a circle.

