# OpenReview forum: "Highlight: Learning Visual Prompts for Vision-Language Models"
_ICLR.cc/2025/Conference — ICLR 2025 Conference Withdrawn Submission_

### Official Review · Reviewer_MdSE · 2024-10-29

**Soundness:** 3
**Presentation:** 3
**Contribution:** 3
**Rating:** 5
**Confidence:** 4

**Summary:**

1.The authors present Highlight, a method that learns visual prompts for Vision-Language Models (VLMs) to improve their ability to localize specific image regions. The key innovation is automatically learning both the shape and color of visual markers in a differentiable manner, rather than relying on manually designed prompts like red circles.

2.The method can be trained either with supervision from text-image region pairs, or without supervision using synthetic captions or images alone.

3.The authors evaluate their approach extensively on RefCOCO datasets using different CLIP models, demonstrating that Highlight outperforms existing visual prompting methods and even ensemble approaches while being more computationally efficient.

**Strengths:**

1.The paper is clear and easy to follow.

2.The research goal is meaningful. They propose to automatically learn visual prompts instead of designing them manually, which is a clever solution to a common problem in vision-language models.

3. Their experimental results are impressive. The method works well across different versions of CLIP models and is more efficient than ensemble methods.

**Weaknesses:**

1.Robustness and Generalizability

- The experiments are conducted exclusively on CLIP variants
- I would like to see evaluations on other popular VLMs (e.g., Flamingo, GPT-4V)
- Without broader validation, it's difficult to assess if this method is truly model-agnostic
- I strongly suggest including at least 2-3 different families of VLMs in the evaluation

2. Theoretical Understanding
- The paper lacks insights into why learned prompts outperform manual ones
- The visualizations don't effectively explain the mechanism's advantages
- I recommend:
(1) Providing case studies of success and failure cases
(2) Including analysis of the learned prompt patterns

3. Image Quality Impact
- Looking at Figure 1, I'm concerned about the image blur introduced by the method
- The authors should address:
(1) How this blur affects the model's performance
(2) If this is a limitation for certain types of images or tasks

**Questions:**

1. I noticed the experiments were only conducted on CLIP models. Have you considered testing the method on other vision-language models? If not, what were the main challenges preventing such evaluations?

2. From Figure 1, I can see that your method introduces some blur to the original images. Could you clarify:
   * Does this blur affect the model's performance?
   * Is this blur necessary for the method to work effectively?

3. Your results show that learned prompts perform better than manual ones, but the mechanism behind this improvement is not clearly explained. Could you:
   * Provide more analysis on why learned prompts work better?
   * Share some failure cases to help understand the method's limitations?

4. I'm interested in the robustness of your method:
   * How stable is the performance across different runs?
   * How sensitive is it to hyperparameter choices?

---

### Official Review · Reviewer_zxHr · 2024-11-02

**Soundness:** 2
**Presentation:** 2
**Contribution:** 2
**Rating:** 5
**Confidence:** 5

**Summary:**

1. [Summary]
This paper focuses on visual prompt learning for vision-language models. It introduces Highlight, a method to automatically learn a visual prompt that highlights regions for a VLM. The proposed method could work in both supervised and unsupervised manner. Experiments show that the proposed method achieves good performances.

**Strengths:**

2. [Strengths]
- a. It introduces Highlight, a method to automatically learn a visual prompt that highlights regions for a VLM.
- b. The experiments are extensive, and the performance of the proposed method is promising.
- c. The method is flexible, which can work in both supervised and unsupervised cases.

**Weaknesses:**

3. [Weaknesses]
1. What is the different between this paper and the other popular visual prompt learning like [a, b]. [a, b] learn tokens as visual prompts. It would be better to discuss and clarify the similarity and difference between this paper and [a, b] and include the discussion in Section Related Work.
[a] Visual Prompt Tuning.
[b] Progressive visual prompt learning with contrastive feature re-formation.

2. The core of this paper is to learn visual prompts for vision-language models. Have the authors tested the proposed method on other VLMs in addition to CLIP? Besides, the Section Related Work lacks a paragraph for introducing VLM. It would be better to add a new paragraph to introduce related work and recent advances in vision-language models, and could also cite the VLM survey [c] as the reference for the readers to know more about VLMs.
[c] Vision-Language Models for Vision Tasks: A Survey
3. Another question is that whether the learnt visual prompts are independent or dependent to input images? They are the same for all input images? Or works differently for different input images?

**Questions:**

as shown in Weaknesses

---

### Official Review · Reviewer_E2fH · 2024-11-03

**Soundness:** 3
**Presentation:** 2
**Contribution:** 1
**Rating:** 3
**Confidence:** 5

**Summary:**

The paper works on a large Vision-Language Model (VLM), especially CLIP. It learns to generate markings on images so that a textual expression related to the image has high correspondence with the image region inside the marking. In short, the authors address the Referring Expression Comprehension (REC) problem using VLMs. While the problem of REC using VLMs like CLIP is not new, unlike previous studies, this paper does not assume the existence of predefined markers like red circle, square or arrow etc. Rather, the authors propose to learn the particular form of markers in both supervised and unsupervised manner. In supervised approach an infoNCE loss is minimized between annotated regions with expressions and the network generated regions (called highlights). In unsupervised approach, the authors use synthetic captions instead of manually annotated descriptions of the crops. Experiments show comparisons with related works that does not use supervision but ensemble of backbones in CLIP.

**Strengths:**

1.	The paper presents a learnable way to predict both shape and color of a visual marker such that regions bounded by such markers in images can correspond to textual descriptions.
2.	The discussion on the relevant works is good.
3.	The use of image crops and synthetic captions for unsupervised training is interesting.

**Weaknesses:**

1.	The novelty of the approach is limited. Let us take the example of the first work in this line which is RedCircle (Shtedritski et al., 2023). The authors there have done extensive analysis on drawing different types of visual prompting e.g., circle, square, arrow etc. with different colors on images and then processing it and the text query via a pretrained VLM. This paper concludes and shows that a red circle works best for REC with CLIP. One may argue that redcircle is dependent on manually tuning the markers for their shapes and colors. Manual tuning has already been replaced by learning to tune the visual markers in Xie et al. (2024a). Xie et al. also did not use supervision for training. The proposed work, on the other hand works good only when supervised training is used (compare row ‘Highlight (unsupervised)’ and ‘RedCircle’ rows as well as ‘Highlight (Supervised)’ and ‘RedCircle’ rows in table 2). The fact that ‘Highlight’ needs to use manual annotation from reference comprehension datasets to perform well, adds to novelty but takes away the major strength and flexibility of VLMs – i.e., zero-shot task transfer. The other incremental novelty of using synthetic texts does not perform well compared to the state-of-the arts.
2.	The closest work to the proposed approach is Xie et al. (2024a) in flavor as this paper also learns to draw a visual marker in zero-shot manner. However, this has not been compared with in the zero-shot setting.
3.	The proposed approach shows its usefulness only in one of the three tasks RedCircle (Shtedritski et al., 2023) evaluated themselves on. How does it perform on the other two tasks namely – ‘Naming Keypoints’ and ‘Keypoint Localization’ is missing.
4.	There are a few strong statements that is not supported well. For example, Line 047 says - alt texts in pretraining image-text datasets summarise global image descriptions. This is in contrast to the common wisdom as alt-texts are usually very short and sometimes cryptic descriptions of a scene. Is there any reference or some examples that you can point to? Similarly, Line 60 says that whether a red circle is the prompt that best elicits these emergent behaviors in VLMs is unclear. However, in Shtedritski et al. (2023), there has been extensive studies on different types of markers e.g., circle, arrow, square, cross etc. or different colors of them. Table 2 in the main paper and table 10 and figure 10 in the supplementary clearly shows red circle works best.
5.	Minor typos: Line 63: ‘unfeasible’ -> ‘infeasible’; Line 261: ‘bounding’ -> ‘bounding box’.

**Questions:**

1.	This is in reference to the point 1 in the weaknesses. The apples-to-apples comparison (unsupervised regime) of the proposed approach with the existing ones can not cement the proposed approach’s betterness. Any comment on this would be good to have.
2.	Comparison with the closest paper Xie et al. (2024a) is missing and in the same level playing ground (without using manual supervision to train) Xie et al. (2024a) works better. Any comment on this would be good to have.
3.	This is related to the third point in the weaknesses. Is it possible to comment on the missing experiments mentioned there?
4.	The strong unsupported claims as detailed in point 4 in ‘weaknesses’ needs to support.
5.	In Table 1, the results are impressive. However, I would like to see, for completeness, what would be the performance of the proposed method if it is measured in the same level playing field i.e., it is unsupervised and uses ensemble of the same backbones as the rest of the competing methods.
6.	I am not comfortable with categorizing methods like RedCircle with sup=x. Theres a fine line between an 'unsupervised' approach and a zero-shot approach. RedCircle can best be described as zero-shot and not 'sup=x' (which means it gets training but no supervision, which is not true for redcircle as it does not get any training).
7.	In figure 2, When both are vision encoders, what encoders are you using? Are the two vision encoders the same in such a case?
8.	The next question is related to eqn. (2). What does this symbol a cross inside circle mean? What is the relation between the two $\mathcal{F}(v)$'s and $I(v)$? What is the difference between upper case I and lower case i in eqn. (2)?

---

> ### Comment · Reviewer_E2fH · 2024-12-03
> **Lack of response**
>
> Without the response from the reviewers, I shall keep my original rating for the submission.

---

### Official Review · Reviewer_sxoN · 2024-11-04

**Soundness:** 3
**Presentation:** 3
**Contribution:** 2
**Rating:** 5
**Confidence:** 4

**Summary:**

This paper improves upon methods that add a manual highlight over image to improve clip performance. The authors optimize a learnable visual prompt and show that they can start from manual annotations to eventually propose the final method which relies on image crops from a pretrained object detector. The authors show that their formulation improves upon previous methods.

**Strengths:**

1. The paper is easy to read and follow.
2. The authors have a nice experimental framework that starts from manual annotations and builds up to a "unsupervised approach".
3. The performance gains over previous methods are non trivial.

**Weaknesses:**

1. The authors have not validated their initial hypothesis that different models will have different biases. Table 6 compares several CLIP models but I am more interested to know if there is any difference in this behavior based on pretraining objective and dataset. How would this compare for SigLIP, SILC, EVA-CLIP, MetaCLIP etc and models trained on OpenAI dataset, Webli, datacomp, etc.
2. How does the method compare with different localization methods. The authors have used MAttNet. Is there data leakage in the localization network? Is it possible to use a generic model here?

**Questions:**

Please see above.

---

### Note · Authors · 2024-12-03

I have read and agree with the venue's withdrawal policy on behalf of myself and my co-authors.